# Residual Effects of Transgenic Cotton on the Intestinal Microbiota of *Dysdercus concinnus*

**DOI:** 10.3390/microorganisms11020261

**Published:** 2023-01-19

**Authors:** Javier Pérez-López, Valeria Alavez, René Cerritos, Guadalupe Andraca-Gómez, Juan Fornoni, Ana Wegier

**Affiliations:** 1Genética de la Conservación, Jardín Botánico, Instituto de Biología, Universidad Nacional Autónoma de Mexico, Avenue Universidad 3000, Circuito Escolar s/n, Ciudad de México 04510, Mexico; 2Posgrado en Ciencias Biológicas, Universidad Nacional Autónoma de México, Avenue Universidad 3000, Circuito Escolar s/n, Ciudad de México 04510, Mexico; 3División de Investigación, Facultad de Medicina, Universidad Nacional Autónoma de México, Avenue Universidad 3000, Circuito Escolar s/n, Ciudad de México 04510, Mexico; 4Instituto de Ecología, Universidad Nacional Autónoma de México, AP 70-275, Ciudad Universitaria, Coyoacán, Ciudad de México 04510, Mexico

**Keywords:** microbiota, insect gut, wild cotton, transgenic plants, networks *cry1ab/ac*, introgression

## Abstract

The interaction among plants, insects, and microbes (PIM) is a determinant factor for the assembly and functioning of natural and anthropic ecosystems. In agroecosystems, the relationships among PIM are based on the interacting taxa, environmental conditions, and agricultural management, including genetically modified (GM) organisms. Although evidence for the unintended effects of GM plants on non-target insects is increasingly robust, our knowledge remains limited regarding their impact on gut microbes and their repercussions on the host’s ecology, especially in the wild. In this study, we compared the gut microbial community of *Dysdercus concinnus* bugs collected on wild cotton (*Gossypium hirsutum*), with and without insecticidal transgenes (*cry1ab/ac*), in its center of origin and diversity. By sequencing the V4–V5 region of 16S rRNA, we show differences in the diversity, structure, and topology of *D. concinnus* gut microbial interactions between specimens foraging cotton plants with and without transgenes. Identifying unintended residual effects of genetic engineering in natural ecosystems will provide first-line knowledge for informed decision-making to manage genetic, ecological, and evolutionary resources. Thus, determining which organisms interact with GM plants and how is the first step toward conserving natural ecosystems with evidence of transgenic introgression.

## 1. Introduction

The traditional study of plant–herbivore interactions has excluded their associated microorganisms, keeping hidden their eco-evolutionary influence on the host. However, in recent decades, the hypothesis of herbivores overcoming the defenses of wild and cultivated plants in cooperation with their gut microbiota [1] has been strengthened. This process enables insects to invade new crops and build resistance to products present in the diet, including insecticides [2]. Therefore, the gut microbiota influences host fitness by modulating aspects of the immune system and behavior, and its composition is a determining factor in the relationship stability between the host and its gut microbiota [3].

Advances in massively parallel sequencing and the development of bioinformatics for their analysis are powerful tools to comprehensively understand the functioning between plant, insect, and microorganism (PIM) interactions in ecosystems, and thus improve our ability to predict and contain ecological perturbations of anthropocentric origin. The massive rise of genetically modified (GM) plants aiming to control pests was approved with partial risk assessments by considering only a fraction of their effect on PIMs, ignoring the role of microbes in driving the evolution of resistance to pesticides, which has recently been discussed [4]. Today, it has been revealed that transgenic sequences introduced by gene flow into wild populations affect non-target organisms [5]. Although, we are just beginning to understand the consequences and the extent of these events.

Recent evidence indicates that a diet with *Bt* toxins (*Bacillus thuringiensis*) modifies the gut microbial structure of non-target insects [6] by significantly decreasing the abundance of Enterococcaceae, and Morganellaceae in the midgut, while increasing the abundance of Enterobacteriaceae [6]. In other cases, after exposure to *Bt*, *Serratia* and *Clostridium* change from asymptomatic intestinal symbionts to hemocele pathogens [7]. Likewise, *Bt* causes a shift in dominance from Firmicutes to Proteobacteria, an increase in Enterobacter and Pseudomonas, and an overall reduction in microbial diversity [8]. Thus, changes in gut microbial structure due to *Bt* exposure should be further investigated in crop centers of origin and diversity, where PIMs share a common evolutionary history. For this reason, investigating the consequences of insecticidal transgenes within populations of crop wild relatives will help to quantify their possible residual effects on native insects and their associated microbiota.

Fiber stainer bugs *Dysdercus* (Pyrrhocoridae) are everyday companions to cotton (*Gossypium hirsutum*), nymphs, and adults feed inside the developing bolls deteriorating the seed and fiber value [9]. Upon reaching the reproductive stage, the wing muscles of female *Dysdercus* histolyze, depriving them of flight, while in males, muscle histolysis never occurs, allowing them to fly throughout their lives [10]. Thus, males become facultative migrants, using hunger as a migration signal, while females feed continuously during copulation, which can last three days [11]. Under these circumstances, constitutive feeding and withholding mobility are likely selectively disadvantageous, for example, when interacting with wild cotton with transgenic properties, as females are more confined than males to unfavorable *Bt* hosts.

The effects of *Bt* on insect gut microbiota led us to hypothesize that feeding on transgenic cotton may modify the abundance and gut microbial composition of *Dysdercus*. Therefore, we used an association network approach to compare the gut microbiota of *D. concinnus*. associated with wild cotton, with and without *cry1ab/ac* (i.e., *cry1ab/ac*^(+)^ and *cry1ab/ac*^(−)^, respectively) and further compared microbial differences between males and females given their differential motility. Identifying the residual effects of genetic engineering in natural ecosystems will allow us to generate plans for managing, monitoring, and containing unexpected consequences.

## 2. Materials and Methods

### 2.1. Experimental Design

Mexico is the center of origin and diversity of American cotton, and despite warnings about gene flow [12] since 1996, genetically modified cotton has been planted in the north of the country. In other words, in Mexico wild populations of cotton coexist with transgenic crops. Wild populations in the South Pacific (SP) and North Pacific (NP) have shown evidence of *cry1ab/ac* introgression since at least 2003 [13]. Therefore, both regions allow long-term investigation of possible unintended consequences of the presence of transgenes in the wild. Plant tissue collections were performed on two populations of SP and NP to detect or rule out the presence of transgenes (*cry1ab/ac*, *cry2ab/ac*, and *CP4 EPSPS*) by end-point PCR assays, following the protocol reported by Vázquez-Barrios et al. [5]. We only detected *cry1ab/ac*.

For our experiment, we selected 30 cotton plants divided into two categories according to their genotype: *cry1ab/ac*^(+)^ (*n* = 19) and *cry1ab/ac*^(−)^ (*n* = 11). From these plants, we collected 65 specimens of *D. concinnus* divided into four categories as described in Table 1. Our sample size reflects the conditions found in the field. Wild cotton populations occur in scattered patches that conform to a metapopulation dynamic; thus, patches constantly become extinct and re-colonized [13] affecting their size.

### 2.2. Specimen Dissection and Gut DNA Extraction

To remove bacteria present on their body surface, we washed the collected specimens with lysozyme (hydrolysis catalyst) for five minutes in a vortex [14]. Next, we extracted the midgut with a longitudinal cut from the anterior to the posterior part of the abdomen with previously sterilized scalpels and tweezers and placed them into individual tubes. Finally, we extracted the intestinal DNA using the DNeasy blood and tissue isolation kit according to the manufacturer’s instructions with the following modification: we set the incubation time to 6 h at 75 °C and performed a second incubation step at 55 °C for five minutes before DNA elution [15].

### 2.3. Amplification and Sequencing

We amplified and sequenced the V4–V5 region of the 16S ribosomal RNA (16S rRNA) gene using the Illumina MiSeq instrument (Illumina, Inc., San Diego, CA, USA) at the Integrated Microbiome Resource (IMR) (Dalhousie University, Halifax, Canada) with primers 515FB [GTGYCAGCMGCCGCGGTAA] and 926R [CCGYCAATTYMTTTRAGTTT] [16].

### 2.4. Demultiplexing, Filtering, and Chimera Check

After sequencing, we processed Illumina raw sequences with QIIME v1.9.1 [17]. Then, we performed the demultiplexing, denoising, chimera checking, and filtering steps with the DADA2 pipeline [18]. First, we demultiplexed the sequences using local scripts and paired the reads using join_paired_ends.py with default arguments. Next, we filtered the joined sequences by quality based on two criteria: (i) discarding sequences with one or more Ns, and (ii) keeping sequences with an overall 75% Phred quality scores >20. After these steps, we obtained 511,168 reads (from a total of 1,598,562) in 72 samples. Afterward, we checked for chimeras with the UCHIME2 algorithm implemented in QIIME [17]. Low-quality chimeric sequences and low abundance sequences (less than five reads per OTU) were removed from the analysis. The raw data (paired-end files) are accessible in the NCBI Sequence Read Archive under accession number PRJNA785562.

Demultiplexed and filtered sequences remained clustered into operational taxonomic units (OTUs) using a de novo strategy with a sequence similarity threshold of 97%. The taxonomy for each OTU was assigned using the Greengenes database [19]. We aligned representative sequences to the Greengenes database with PyNAST [19] and constructed a maximum likelihood phylogenetic tree using FastTree2 [20]. Finally, the obtained OTU table was filtered using a minimum cluster size of 0.001% of the reads.

### 2.5. Statistical Analyses of Molecular Data

After cleaning the 16S data, to compare the gut microbial diversity of *D. concinnus* associated with females and males that fed on *cry1ab/ac*^(+)^ and *cry1ab/ac*^(−)^ cotton, we calculated Shannon–Wiener indexes. Statistically significant differences in alpha diversity between microbial communities were calculated with the Wilcoxon rank sum test using the “vegan” package in R [21]. To compare gut microbial diversity in the interaction between host sex and diet, we performed ANOVA tests. In addition, we built a network of gut microbial associations of females and males that fed on *cry1ab/ac*^(+)^ and *cry1ab/ac*^(−)^ cotton, following Matchado et al. [22]. We used Pearson’s method to calculate positive or negative associations between taxa, the Student’s t-test to determine significant differences, and the *multRepl* method (multiplicative replacement) to efficiently handle zeros. Moreover, we used an adaptive Benjamini–Hochberg model (*adaptBH*) [23] to control the false discovery rate (FDR) of multiple pairwise comparisons (i.e., the probability that the link formed between pairs of nodes is true after being rejected by the statistical test). This method assesses if a particular element is associated with some condition adjusted to a linear model [24], allowing for the proportion of significant tests that lack the association to be estimated. The Benjamini–Hochberg method assumes that all null hypotheses are true when estimating the number of null hypotheses erroneously considered false. Consequently, the FDR estimate is inflated and, therefore, conservative.

To determine substructures (neighborhoods) within the gut microbial networks, we employed a modularity optimization algorithm (fast greedy) [25]. We compared the similarity between clusters with the Adjusted Rand index (ARI). ARI values range from 0 (no agreement) to 1 (perfect agreement) [26]. We constructed a differential network with the taxa shared between females and males fed on *cry1ab/ac*^(+)^ and *cry1ab/ac*^(−)^ cotton and compared the edges with Fisher’s test. The differential association network allows for determination of the change in the associations (from positive to negative, and vice versa) between pairs of nodes given the compositional abundance of the taxa. Finally, we compared the relative compositional abundance of bacterial groups in the intestines of males and females using a Wilcoxon rank sum test, a non-parametric method equivalent to a *t*-test.

All statistical analyses were conducted in the R statistical computing environment [27]. The packages used for the handling, construction, comparison, and visualization of the microbial networks were: *microbiome* [28], *igraph* [29], *qiime2R* [30], *NetCoMi* [31], and *ggpubr* [32]. All codes are available at: https://github.com/conservationgenetics/, accessed on 1 December 2022.

## 3. Results

### 3.1. Dysdercus Concinnus Gut Bacterial Communities

All high-quality reads fell within 16 phyla. The most abundant were: Firmicutes (49%), Proteobacteria (34%), Actinobacteria (14%), and Tenericutes (4%) (Figure 1A). The most abundant bacterial classes in specimens that fed on *cry1ab/ac*^(+)^ cotton were: Bacilli (41%), Gammaproteobacteria (24%), Clostridia (14%), Alphaproteobacteria (10%), and Coriobacteriia (10%); while those on a *cry1ab/ac*^(−)^ diet were: Bacilli (31%), Alphaproteobacteria (24%), Coriobacteriia (13%), Gammaproteobacteria (13%), Clostridia (11%), and Mollicutes (7%) (Figure 1B).

When comparing between the sexes, the most abundant families in females that fed on *cry1ab/ac*^(+)^ cotton were: Enterobacteriaceae (36%), Enterococcaceae (27%), Lachnospiraceae (9%), and Streptococcaceae (9%). On the other hand, we recorded a more equitable intestinal bacterial community in females that fed on *cry1ab/ac*^(−)^ cotton, represented by the families: Bartonellaceae (24%), Lachnospiraceae (12%), Bacillaceae (6%), Coriobacteriaceae (6%), Enterobacteriaceae (6%), Enterococcaceae (6%), Lactobacillaceae (6%), and Streptococcaceae (6%).

Regarding males that fed on wild *cry1ab/ac*^(+)^ cotton, the families with the highest representation were: Enterobacteriaceae (28%), Bartonellaceae (17%), Coriobacteriaceae (17%), Lachnospiraceae (17%), Enterococcaceae (11%), and Streptococcaceae (6%). Whereas in males that fed on *cry1ab/ac*^(−)^ cotton, the most abundant families were: Coriobacteriaceae (21%), Bartonellaceae (18%), Enterobacteriaceae (18%), Streptococcaceae (14%), Lachnospiraceae (11%), and Enterococcaceae (7%).

We recorded lower diversity of gut microbes in *D. concinnus* collected on *cry1ab/ac*^(+)^ cotton (Figure 2A and Table 2). Differences in microbial diversity given diet–sex–host interaction were not significant, except in the females, where the lowest microbial intestinal diversity was recorded in those fed with *cry1ab/ac*^(+)^ cotton, although this difference was marginally significant (Table 2). In addition, we found a greater intestinal microbial dominance in *D. concinnus* that fed on *cry1ab/ac*^(+)^ cotton, as they presented a lower proportion of non-core species (Figure 2B).

Assessing the genus-level diversity in bugs that fed on cotton with transgenes through a Wilcoxon test, we registered a greater abundance of *Blautia* (Firmicutes: Clostridia), *Klebsiella* (Proteobacteria: Gammaproteobacteria), *Bacteroides* (Bacteroidetes: Bacteroidia), Isosphaeraceae (Planctomycetes: Planctomycetia), Lentisphaeraceae (Lentisphaerae: Lentisphaeria), and *Planctomyces* (Planctomycetes: Planctomycetia) in females, and *Coriobacterium* (Actinobacteria: Coriobacteriia), *Clostridium* (Firmicutes: Clostridia), *Bacteroides* (Bacteroidetes: Bacteroidia), *Pantoea* (Proteobacteria: Gammaproteobacteria), and *Serratia* (Proteobacteria: Gammaproteobacteria) in males (Table 3).

### 3.2. Bacterial Community Networks in the D. concinnus Gut

The presence of *cry1ab/ac*^(+)^ in the *D. concinnus* diet appears to influence the intestinal microbial structure of males and females differently. We found significant differences in the topology of the intestinal microbial network of females feeding on *cry1ab/ac*^(+)^ cotton compared to those feeding on transgene-free cotton (Figure 3, Table 4). We found a lower degree and eigenvector centrality of central nodes in the network of females feeding on *cry1ab/ac*^(+)^ cotton (Table 4 and Table 5). Likewise, the number of links and identity of the central nodes *(hubs)* also displayed differences (Figure 3A,B, Table 4). In females feeding on *cry1ab/ac*^(−)^ cotton, the taxa with the most significant influence belong to the genera *Clostridium* and *Pseudomonas*, while those feeding on *cry1ab/ac*^(+)^ cotton belong to the genera *Clostridium* and *Faecalibacterium* (Table 4). In addition, we found that in *cry1ab/ac*^(+)^-fed females, bacteria such as *Bifidobacterium*, *Ruminococcus*, *Pseudoramibacter*, and *Clostridium* present a positive association. In contrast, we observed a negative association in females on a transgene-free diet (Figure 3A).

In males, we did not find significant differences in the topological properties of the intestinal microbes’ network associated with their diet (Table 4 and Table 5). However, they differ in the hubs that determine network structure. In *cry1ab/ac*^(+)^-fed males, the most significant taxa in terms of network structure belong to the genera *Clostridium*, *Dysgonomonas*, and *Serratia*, whereas for males under a *cry1ab/ac*-free diet, they are *Clostridium* and *Bacillus* (Figure 3C,D and Table 4).

From the differential association network between males fed with and without *cry1ab/ac*, we observed more significant heterogeneity in the change from positive to negative associations and vice versa than in females (Figure 4A,B). For example, Figure 4B shows through golden edges the negative associations in males feeding on *cry1ab/ac*^(−)^ cotton that is positive in *cry1ab/ac*^(+)^-fed bugs. Conversely, the purple edges exhibit positive associations in males feeding on transgene-free cotton that are negative in males on a *cry1ab/ac*^(+)^ diet. The ARI index indicates that the network substructures of females that were fed on cotton with and without *cry1ab/ac* are different (Table 5). In contrast, we did not find significant differences when comparing the neighborhoods between the males.

## 4. Discussion and Conclusions

Crop wild relatives are critical to ensuring global food security [33]. However, they are threatened by various causes, including the release of genetically modified organisms [34]. In this study, we identified and analyzed one residual effect of transgenic flow in the center of origin and diversity of American cotton. Our results suggest a decrease in microbial diversity gut and changes in insect–microbe interactions when *cry1ab/ac* is present in a phytophagous insect’s diet. First, we will discuss the key bacterial groups found in this study and their physiological role in *Dysdercus*. Then, we will address the influence of these bacteria on their host’s fitness, and we will share some thoughts on the ecological and evolutionary consequences of our findings. Even with a limited number of plants, we detected results due to the genetic expression of transgenes in a natural population, where plants coexist with their symbionts and antagonists. These results should be considered the first line of evidence requiring further investigation with a larger sample size.

### 4.1. Key Intestinal Microbial Taxa in Dysdercus Species and Their Influence on Fitness Traits

The gut microbiota depends on the diet and the physiological state of the host [35]. It is a genetic reservoir with evolutionary potential that allows the host to cope with contrasting environmental conditions. This evidence supports the assumption that the gut microbiota has influenced the evolutionary history of the genus *Dysdercus* [36,37]. Consequently, any change in the structure of the intestinal microbial community should be of interest to understand the ecological and evolutionary implications on its host insect.

Our study recovered three of the phyla (Actinobacteria, Firmicutes, and Proteobacteria) previously characterized as the dominant groups in the intestinal microbiota of other *Dysdercus* species [38] and within the family Pyrrhocoridae [39]. However, we observed changes in the abundance of some bacterial groups in hosts feeding on *cry1ab/ac*^(+)^ cotton. For instance, we found a greater abundance of *Blautia* (Firmicutes: Clostridia) in females. *Blautia* is associated with the degradation of complex polysaccharides into short-chain fatty acids (e.g., acetate, butyrate, and propionate) that play a role as substrates for energy production [40]. In other insects, the production of short-chain fatty acids by interacting with vitellogenin promotes the host’s growth and increases its sensitivity to sugar [41].

A greater abundance of *Blautia* in *D. concinnus* feeding on *cry1ab/ac*^(+)^ cotton may represent a higher number of short-chain fatty acids leading to a higher growth rate and probably a greater reproduction rate by reaching sexual maturity earlier. Therefore, gut microbes such as *Blautia* can modify the host’s phenotype by enhancing its nutritional capacity, directly influencing its fitness [42]. Functional similarities of *Blautia* are reported for the mammalian gut [43]. Perhaps it is not surprising to find functional similarities between mammals and insects because the latter’s microbial communities maintain an oxic–anoxic gradient in the intestine with reduced pH and redox potential [44]. However, the degree to which similar mechanisms underlie the observed similarities of gut microbiota in dissimilar hosts needs further investigation.

Moreover, we observed a higher abundance of *Coriobacterium* (Actinobacteria: Coriobacteriia) in *D. concinnus* males feeding on *cry1ab/ac*^(+)^ cotton (Table 3). This bacterial genus is a major component of the structure of microbial networks in both males and females feeding on transgenes. In other *Dysdercus* species, the abundance of *Coriobacterium* is associated with higher growth rates, greater reproductive success, and lower mortality [45], suggesting that symbiotic actinobacteria play an essential role in their host’s life history. Therefore, the transgenic cotton diet could improve the fitness of *D. concinnus* by directly or indirectly influencing the intestinal abundance of *Coriobacterium*, particularly in males, promoting the mating between insects with enriched concentrations of the symbiont. Although the metabolic importance of *Coriobacterium* for *Dysdercus* and other representatives of the family Pyrrhocoridae has been proven [46], experimental corroboration is required. Thus, more ecological and evolutionary research is needed to unravel the microbial mechanisms interacting with host metabolism to respond to challenging environments.

In addition to being nutritionally necessary, *Dysdercus* gut bacteria complement the host’s immune system by preventing parasite invasions [37]. Thus, a stable and consistent gut microbial community significantly contributes to insect fitness, suggesting that both host and symbionts have evolved concurrently for millions of years [47]. Our study recorded a higher abundance of *Bacteroides* (*Bacteroidetes*: *Bacteroidia*) in hosts feeding on *cry1ab/ac*^(+)^ cotton (Table 3). *Bacteroidetes* reduce oxygenation levels in the intestine [48], allowing strict and facultative anaerobic bacteria to establish and grow by generating a hospitable microenvironment, which ultimately modifies the intestinal microbial structure and composition by enabling or restricting the colonization of new microbes. For example, some anaerobic groups such as *Clostridium* and *Klebsiella* sp. could benefit from these intestinal conditions and proliferate. Interestingly, we found both groups in greater abundance in females feeding on *cry1ab/ac*^(+)^ cotton (Table 3). From this perspective, the diet with transgenes could modify the microenvironmental conditions in the intestine and alter the microbe–microbe relationships in *D. concinnus*.

### 4.2. What Does Not Kill You Makes You Stronger (Inside and Outside the Gut Microbiota): Ecological and Evolutionary Considerations

The interactions of gut microbes that enhance the fitness of their hosts reinforce the hypothesis of adaptive diversification as a process strongly linked to the joint expression of PIM phenotypes [49,50]. However, insect–host genetic variation could facilitate the preferential association of specific microbes for specific environmental conditions [3], facilitating invasions into new niches [51]. Therefore, genotype (insect host) × genotype (microbe) × environment interactions need to be incorporated into future research to fully understand the mechanisms underlying the transgenic plants’ interactions with natural enemies in wild populations.

Our findings strongly suggest that the expression of *cry1ab/ac* in wild cotton plants modified the structure of the gut microbiota of *D. concinnus*. In *D. concinnus* females fed on *cry1ab/ac*^(+)^ cotton, the most significant hubs for network composition were taxonomically different and established fewer links with other microbes than the other groups assessed in this study. Furthermore, our results suggest that the *cry1ab/ac*^(+)^ cotton diet changed microbial antagonistic associations towards cooperation (Figure 3) and decreased global microbial diversity (Figure 2A). These residual effects could compromise bacterial relationships with the host since their interaction stability depends on microbe–microbe associations in the gut [52]. More work is needed to assess the genotype × genotype × environment interactions involved in this system; however, documenting the residual effect of *cry1ab/ac* on *D. concinnus* microbiota is the first step toward that goal.

The ecological and evolutionary influence of the microbiome on the host depends on the heritability of gut microbes to progeny. In the genus *Dysdercus*, the microbiota’s transmission to its offspring is highly stable. Inheritance is possible transovarially or through coprophagia and cannibalism [46]. Thus, mixed transmission (vertical and horizontal) of gut microbes could rapidly spread successful phenotypes within populations and thus facilitate the rise of emerging pests or improve the fitness of existing ones. The introduction of transgenic cotton to China’s Shandong and Hebei provinces teaches a powerful lesson on the spread of gut microbes in pest insect populations. With experimental and field evidence, Itoh et al. [52] noted that exposure to transgenic cotton crops facilitated beneficial interactions between the target pest (*Helicoverpa armigera*) and associated microbes (densovirus [HaDV2]) that improve host fitness in transgenic crops, suggesting residual effects of plant transgenes on insects and their associated microbes. For this reason, the GMO diet could drive the evolutionary trajectory of insects, especially in transgenic-rich environments such as crop fields. However, although there are some trends regarding the effect of transgenes on non-target insects, our knowledge of the ecology of transgenic plants in wild ecosystems is still superficial.

Modifying phenotypes affects the populations’ variation subjected to natural selection [53]. In this sense, the change in the community structure of the intestinal microbiota of *D. concinnus* could influence the evolutionary history of its hosts. The phenotypic differences presented by plants with transgenes and their influence on determining elements for the intestinal microbial structure could represent a new habitat for associated insects by offering different resources and conditions [54]. Thus, the presence of *cry1ab/ac* in wild cotton could favor the appearance of emerging pests by influencing intestinal microbial relationships that enhance their survival and reproduction. The higher the selection pressure in time and space, the faster the evolutionary response of pests [55]. The endogenous and constitutive expression of transgenes in wild cotton represents continuous exposure to this selection pressure, which can dramatically accelerate resistance evolution [56]. Consequently, the insecticidal resistance mediated by gut microbes could nullify the insecticidal property of *Bt* [57]. This problem would place crops and protocols safeguarding the world’s food security in a vulnerable position. For instance, management and containment plans for agricultural pests would be rendered obsolete, presenting worldwide challenges in the face of insects’ resistance evolution [56]. Furthermore, with the advent of transgenic introgression in the wild, the consequences of differential insect fitness deserve special attention because the conservation of crop wild relatives is a priority within centers of origin and diversity. We do not yet know the full implications of altering PIM interactions within these settings.

Several academic papers have documented the ecological and evolutionary changes that could gradually intensify by the presence of transgenes [5,58]. The consequences on wild cotton have not yet been integrated; for example, the altered defense system of cotton plants with *cry* transgenes that continually secrete nectar [5] interacts with *D. concinnus* females and males by sharing the same host plant. Therefore, it is necessary to study the interaction between these arthropods. Conducting studies on this system in the field is significantly complicated; therefore, an infinite list of research possibilities that require multiple resources could be planned. However, we suggest that the cause of the observed ecological changes (i.e., transgenic introgression) is recognizable with a relatively simple molecular analysis, so it would be ideal to discuss opportunities to mitigate the presence of transgenes to prevent complex consequences.

Furthermore, antibiotic resistance genes are used in pre-plant transformation and during transformation to select bacteria, cells, and plants with stable properties [59]. Transgenic cotton’s most common antibiotic resistance genes are *aad* (neomycin and kanamycin) and *ntpII* (aminoglycoside antibiotics such as spectinomycin and streptomycin); thus, all releases of transgenic cotton into the environment have at least one of these markers. While this may influence the differences found in this study, it also represents an additional source of concern at more than one trophic level. In *D. concinnus*, gut microbes already survive the constitutive presence of antibiotics in GM cotton. Other residual effects may emerge in the future as immature cotton bolls (which may contain juvenile *Dysdercus* inside) are used constantly to feed cattle (which have their own load of antibiotics). These factors could promote the spread of bacteria resistant to these antibiotics and others residual effects on cattle gut microbiomes. Moreover, although gene transfer among bacteria is more likely than between plants and the gut microbiome, the latter should not be considered impossible [60].

In summary, our results suggest that the expression of *cry1ab/ac* in wild cotton within its center of origin and diversity has residual effects in the intestinal microbiota of *D. concinnus* in both males and females. We discovered that changes in intestinal microbial elements in females feeding on *cry1ab/ac*^(+)^ cotton are associated with fitness traits that have been key throughout *Dysdercus’* ecological and evolutionary history. Thus, insect evolution cannot be understood without considering their associated microorganisms. This study is valuable for identifying a silent threat to cotton’s wild relatives and landraces. The changes in the intestinal microbial network reported in this study will contribute to understanding the intestinal microbial dynamics of pest insects and their ecological and evolutionary influence on their hosts.

## Figures and Tables

**Figure 1 microorganisms-11-00261-f001:**
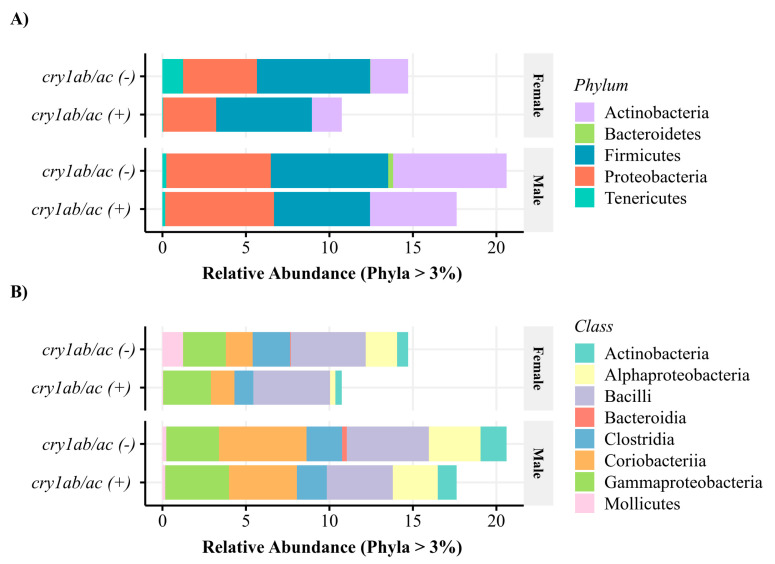
Relative composition of the intestinal microbiota of male and female *D. concinnus* feeding on wild cotton with and without *cry1ab/ac* at the level of (**A**) phyla and (**B**) class.

**Figure 2 microorganisms-11-00261-f002:**
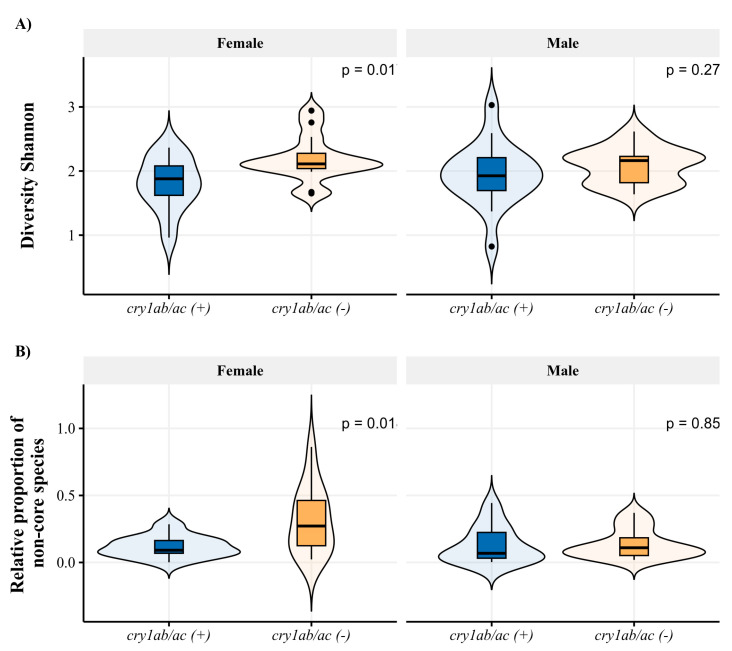
(**A**) Microbial diversity and (**B**) relative proportion of non-core species of the intestinal microbiota of females and males of *D. concinnus* feeding on wild cotton with and without *cry1ab/ac*.

**Figure 3 microorganisms-11-00261-f003:**
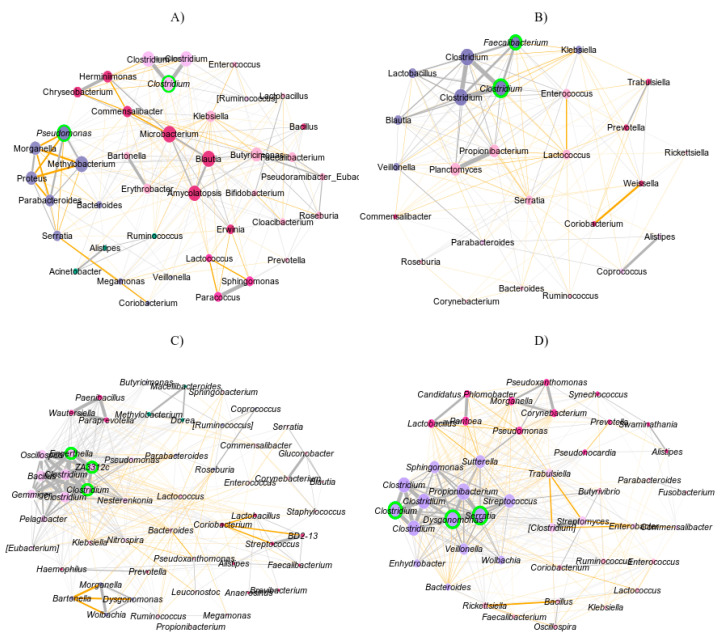
Association network of the intestinal microbiota of *D. concinnus* (**A**) Females feeding on *cry1ab/ac*^(−)^ cotton; (**B**) females feeding on *cry1ab/ac*^(+)^ cotton; (**C**) males feeding on *cry1ab/ac*^(−)^ cotton; and (**D**) males feeding on *cry1ab/ac*^(+)^ cotton. The color of the nodes represents their membership in a neighborhood. The size of the node is proportional to centrality per eigenvector. Nodes identified as hubs are highlighted in green. Negative and positive associations between pairs of nodes are shown in gray and orange, respectively.

**Figure 4 microorganisms-11-00261-f004:**
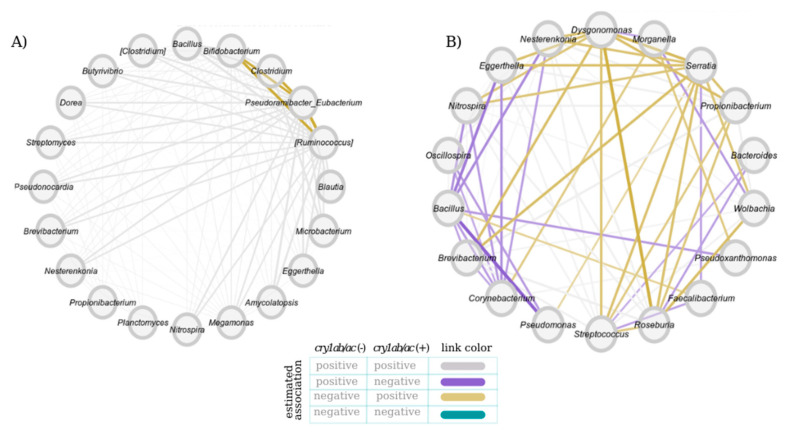
Differential network of the intestinal microbiota of (**A**) females and (**B**) males feeding on *cry1ab/ac*^(+)^ and *cry1ab/ac*^(–)^ cotton. Gray and green links represent congruent associations (positive or negative, respectively) between node pairs regardless of the diet. In contrast, purple and gold links depict incongruous associations.

**Table 1 microorganisms-11-00261-t001:** Number of *D. concinnus* specimens (n) collected by sex and diet (i.e., the cotton genotype they fed upon).

Sex	Cotton Genotype	*n*
Female	*cry1ab/ac* ^(+)^	11
	*cry1ab/ac* ^(−)^	15
Male	*cry1ab/ac* ^(+)^	18
	*cry1ab/ac* ^(−)^	21
Total	-	65

**Table 2 microorganisms-11-00261-t002:** ANOVA test results show significant differences in the diversity and richness of non-core microbes associated with the gut of *Dysdercus* as a function of sex and diet with and without transgenes.

Variable	Factor	df	Mean Sq	F Value	*p*-Value
Diversity	Diet	1	0.89	6.15	0.01
	sex	1	0.001	0.01	0.92
	Interaction	1	0.23	1.61	0.2
Non-core	Diet	1	0.11	4.93	0.03
	sex	1	0.14	5.9	0.01
	Interaction	1	0.13	5.73	0.01

**Table 3 microorganisms-11-00261-t003:** Wilcoxon test results showing the abundance differences in intestinal bacteria between males and females of *D. concinnus* collected on *cry1ab/ac*^(+)^ and *cry1ab/ac*^(−)^ cotton.

Sex	Phylum	Class	Order	Family	Genus	Species	*p* Value
Female	Bacteroidetes	Bacteroidia	Bacteroidales	Bacteroidaceae	*Bacteroides*	*ovatus*	0.04
	Bacteroidetes	Bacteroidia	Bacteroidales	Bacteroidaceae	*Bacteroides*	*ovatus*	0.05
	Firmicutes	Bacilli	Lactobacillales	Enterococcaceae	*Enterococcus*	-	0.04
	Firmicutes	Clostridia	Clostridiales	Lachnospiraceae	*Blautia*	-	0.05
	Firmicutes	Clostridia	Clostridiales	-	-	-	0.04
	Lentisphaerae	[Lentisphaeria]	Lentisphaerales	Lentisphaeraceae	-	-	0.04
	Planctomycetes	Planctomycetia	Gemmatales	Isosphaeraceae	-	-	0.04
	Planctomycetes	Planctomycetia	Planctomycetales	Planctomycetaceae	*Planctomyces*	-	0.04
	Proteobacteria	Gammaproteobacteria	Alteromonadales	OM60	-	-	0.04
	Proteobacteria	Gammaproteobacteria	Enterobacteriales	Enterobacteriaceae	*Klebsiella*	*oxytoca*	0.04
	Proteobacteria	Gammaproteobacteria	Enterobacteriales	Enterobacteriaceae	-	-	0.04
	Proteobacteria	Gammaproteobacteria	Enterobacteriales	Enterobacteriaceae	-	-	0.06
Male	Actinobacteria	Coriobacteriia	Coriobacteriales	Coriobacteriaceae	*Coriobacterium*	-	0.03
	Bacteroidetes	Bacteroidia	Bacteroidales	Bacteroidaceae	*Bacteroides*	*ovatus*	0.05
	Cyanobacteria	Synechococcophycideae	Synechococcales	Synechococcaceae	*Synechococcus*	-	0.05
	Firmicutes	Bacilli	Lactobacillales	Enterococcaceae	*Enterococcus*	-	0.01
	Firmicutes	Clostridia	Clostridiales	Lachnospiraceae	*Clostridium*	*hathewayi*	0.05
	Proteobacteria	Gammaproteobacteria	Enterobacteriales	Enterobacteriaceae	-	-	0.02
	Proteobacteria	Gammaproteobacteria	Enterobacteriales	Enterobacteriaceae	-	-	0.05
	Proteobacteria	Gammaproteobacteria	Enterobacteriales	Enterobacteriaceae	-	-	0.05
	Proteobacteria	Gammaproteobacteria	Enterobacteriales	Enterobacteriaceae	*Pantoea*	-	0.05
	Proteobacteria	Gammaproteobacteria	Enterobacteriales	Enterobacteriaceae	*Serratia*	-	0.05
	Proteobacteria	Gammaproteobacteria	Enterobacteriales	Enterobacteriaceae	*Trabulsiella*	*farmeri*	0.05
	Proteobacteria	Gammaproteobacteria	Enterobacteriales	Enterobacteriaceae	*Trabulsiella*	-	0.05
	Proteobacteria	Gammaproteobacteria	-	-	-	-	0.05

**Table 4 microorganisms-11-00261-t004:** Structural properties of the intestinal microbial networks of male and female *D. concinnus* collected on *cry1ab/ac*^(+)^ and *cry1ab/ac*^(−)^ cotton.

Properties	Female		Male	
	*cry1ab/ac*^(−)^ Cotton	*cry1ab/ac*^(+)^ Cotton	*cry1ab/ac*^(−)^ Cotton	*cry1ab/ac*^(+)^ Cotton
Clustering coefficient	0.57	0.6	0.64	0.65
Modularity	0.19	0.15	0.17	0.14
Positive edge percentage	58.33	53.6	68.24	57.06
Edge density	0.27	0.42	0.23	0.41
Natural connectivity	0.03	0.08	0.06	0.07
Vertex connectivity	3	3	1	2
Edge connectivity	3	3	1	2
Average dissimilarity	0.95	0.91	0.95	0.92
Average path length	1.59	1.43	1.81	1.45
Hubs	*Clostridium* and *Pseudomonas*	*Clostridium* and *Faecalibacterium*	*Clostridium* and *Bacillus*	*Clostridium*, *Dysgonomonas*, *and Serratia*
Number of clusters	5	3	5	4

**Table 5 microorganisms-11-00261-t005:** Jaccard indexes comparing the similarity between intestinal microbial network nodes with the highest centrality per eigenvector of female and male *Dysdercus* collected on *cry1ab/ac*^(+)^ and *cry1ab/ac*^(−)^ cotton. Significant differences are shown in bold.

Sex	Properties	Jacc	*p* (≤Jacc)	*p* (≥Jacc)
Female	Degree	0.55	0.98	0.03
	Betweenness centrality	1	1	0.33
	Closeness centrality	0.16	0.18	0.94
	Eigenvector centrality	0.55	0.98	0.03
	ARI	0.28	-	0.02
Male	Degree	0.21	0.26	0.89
	Betweenness centrality	0	1	1
	Closeness centrality	0.12	0.19	0.96
	Eigenvector centrality	0.21	0.26	0.89
	ARI	0	-	1

## Data Availability

The raw data (paired-end files) are accessible in the NCBI Sequence Read Archive under accession number PRJNA785562. All codes are available at https://github.com/conservationgenetics/, accessed on 1 December 2022.

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
