# Peer review of "Residual Effects of Transgenic Cotton on the Intestinal Microbiota of Dysdercus concinnus"

_microorganisms, 2023, doi:10.3390/microorganisms11020261_

Round 1

Reviewer 1 Report

This paper has been submitted to Microorganisms and presents the results from a study on gene flow from genetically modified organisms into a plant-insect-microorganism (PIM) ecosystem. The main goal was to detect residual effects of transgenic cotton on the intestinal microbiota of Dysdercus (Hemiptera). It was compared in bioassays the gut microbial community of Dysdercus bugs collected on wild cotton with and without insecticidal transgenes. Thus, bugs were divided into four categories: bugs (male or female) collected in wild populations of cotton with or without transgenes. Insects were split in gender as females’ wing muscles histolyze upon reaching the reproductive stage which does not occur in males; thus, males are more mobile within and in between habitats than female, which could affect the contact with transgenes from wild cotton. It was found differences in the diversity, structure, and topology of Dysdercus gut microbial, with a decrease in gut microbial diversity and changes in insect-microbe interactions when transgenes are present in the insect's diet. It is discussed the role of microbial community found and their physiological implications for Dysdercus, as well as their influence on host’s fitness. It is speculated that changes in intestinal microbial community in females feeding on cotton containing transgenes are associated with fitness traits. The study is well written, but it lacks a clear question and hypothesis. All the fitness and ecological aspects raised by the authors in the study are speculative, as they have not experimentally tested none of such hypotheses. Despite the journal scope I missed an in-depth insight into the insect behaviour, biology, physiology or ecology.

Comments:

The study is on the genera Dysdercus or on an specific species? I have just saw the mention on Dysdercus, but none species. This should be very clear!

Abstract: I miss the conclusion and the main results here.

L35: I would suggest put "hidden" after keeping.

L39: Insert dot after [2].

L46: I suggest take off “the impact”.

L48: I think the sentence “ignoring the role of microbes as drivers of resistance evolution” rather vague.

L49-50: I suggest changing “lack neutral effects when interacting with” to “affect”.

L64: According to the nomenclature code it should come as “Dysdercus (Pyrrhocoridae)”

L65: only in Mexico?

L68-70: This information is very important and should be clear what is it relationship with genetic flow of transgenes.

L70-72: How? It is not clear to me how gut microbiota gave Pyrrhocoridae the ability to feed on Malvales and diversify.

L72-74: It is a loose sentence and unconnected with other information.

L71: Why is “Pyrrhocoridae” highlighted from the text?

L91-92: Another loose and unconnected sentence.

L96: What is “PS”?

L217: “Blautia” should not be highlighted from the text?

L283: “gut” should come before “microbial”

Author Response

Response to Reviewer 1 Comments

We are grateful for the comments and suggestions of reviewer 1, which undoubtedly contributed to enrich our manuscript (#2104795). The comments and the authors' response are listed below.

Point 1: L35: I would suggest put "hidden" after keeping.

Response 1: We change the word "hidden" after the word "keeping".

Point 2: L39: Insert dot after [2].

Response 2: We insert a dot after cite [2].

Point 3: L46: I suggest take off “the impact”.

Response 3: We remove “the impact”.

Point 4: L48: I think the sentence “ignoring the role of microbes as drivers of resistance evolution” rather vague.

Response 4: To further elaborate on the meaning of the phrase, we add the quote from Siddiqui, a review article that compiles evidence for the role of insect gut microorganisms in pesticide degradation. We now know that microbes break down various allelochemicals and pesticides, providing insects with numerous opportunities to develop detoxifying symbiotic relationships. The gut microbiota plays multiple roles in host physiology, such as immune modulation and toxin degradation. Arguably, microbiota evolve faster than their insect hosts, resulting in rapid adaptation of pests to pesticides through mutualistic microbes. Moreover, insects can rapidly obtain new metabolic activities and colonize new ecological niches through symbiotic interactions with microbiota that have previously fully evolved well-refined metabolic pathways. Therefore, we mention in our manuscript that approving the use of transgenic plants in agriculture by partial risk assessments without taking into account the associated microbiota is at least a miscalculation.

Point 5: L49-50: I suggest changing “lack neutral effects when interacting with” to “affect”.

Response 5: We replace “lack neutral effects when interacting with” to “affect”.

Point 6: L64: According to the nomenclature code it should come as “Dysdercus (Pyrrhocoridae)”

Response 6: Fixed "Dysdercus" to stick with the name code. Also we rewrite the paragraph of L64-L80.

Point 7: L65: only in Mexico?

Response 7: No, it is not just a native companion of Mexico, the genus Dysdercus is a major pest of commercial cotton crops worldwide. We have deleted the word Mexico. Also we rewrite the paragraph of L64-L80.

Point 8: L68-70: This information is very important and should be clear what is it relationship with genetic flow of transgenes.

Response 8: We agree with this observation, so in order to give importance to this sentence we decided to rewrite the paragraph of line 64-L80.

Point 9: L70-72: How? It is not clear to me how gut microbiota gave Pyrrhocoridae the ability to feed on Malvales and diversify.

Response 9: We have decided to rewrite the paragraph on line 64-L80. But to address the clarification let us recall that evolutionary adaptations to exploit toxic or nutritionally challenging host plants represent a major force driving the diversification of phytophagous insects. In the study by Sudakaran et al., 2015 they investigated whether acquiring vitamin-contributing symbionts allowed their hosts to diversify towards the seeds of nutritionally unbalanced and chemically well-defended malvaceous plants as a food source. Their results indicate that vitamin-contributing Actinobacteria (Coriobacterium and Gordonibacter), as well as Firmicutes (Clostridium) and Proteobacteria (Klebsiella) are widespread in Pyrrhocoridae, but absent in the sister family Largidae and other external taxa. Their phylogenetic dating analyses based on the fossil record revealed an origin of the core microbiota of Pyrrhocoridae in the late Cretaceous (81.2-86.5 million years ago), following the transition from crypt-associated beta-proteobacterial symbionts to an anaerobic community located in the M3 region of the midgut. This change in symbiotic syndromes (i.e., identity and location of the symbionts) and the acquisition of the core microbiota of the pyrrhocorids followed the evolution of their preferred host plants (Malvales), suggesting that the symbionts facilitated the adaptation of their hosts to this imbalanced nutritional resource and allowed subsequent diversification in an ecological niche poor in competition.

Sudakaran, S.; Retz, F.; Kikuchi, Y.; Kost, C.; Kaltenpoth, M. Evolutionary Transition in Symbiotic Syndromes Enabled Diversification of Phytophagous Insects on an Imbalanced Diet. ISME J. 2015, 9, 2587–2604, doi:10.1038/ismej.2015.75.

Point 10: L72-74: It is a loose sentence and unconnected with other information.

Response 10: OK, we have removed the sentence to maintain the flow and connection with the rest of the manuscript.

Point 11: L71: Why is “Pyrrhocoridae” highlighted from the text?

Response 11: OK,this is a formatting error. We have deleted the highlighting of Pyrrhocoridae throughout the manuscript.

Point 12: L91-92: Another loose and unconnected sentence.

Response 12: we rewrite the paragraph of L91-L94

Point 13: L96: What is “PS”?

Response 13: It is the abbreviation for South Pacific (SP) and North Pacific (NP). In the first version of the manuscript the abbreviations were in Spanish. Now we have changed them to English.

Point 14: L217: “Blautia” should not be highlighted from the text?

Response 14: OK, we highlight the bacterial genera and species mentioned throughout the manuscript, including Blautia.

Reviewer 2 Report

The article submitted for review concerns an interesting, little-explored area of the environmental impact of transgenic plants. The article is well written, forming a logical, consistent whole. The data obtained is presented in a clear and readable manner. Among the weaknesses of the article is the limited research material, which does not allow to draw deeper conclusions. In addition, the Dysdercus species used in the study was not determined.

Minor comments:

L22: “evolutionary history” - this is an overreaching statement.

L92: insert a space before GM.

L113: replace “sterile” to “sterilized”.

L144: replace “statistical” to “statistically”.

Figure 1: If possible - change the type of the chart so that it is readable as a black and white printout. Enlarge the fonts.

Table 2: change “Df” to “df”.

Table 3: genus and species names of bacteria should be in italics.

Table 4: insert commas between genus names of bacteria.

L288: remove “exciting, don't judge your own research results.

L288: change “mutualistic” to “symbionts”.

L311: “reproductive age earlier” - age is measurable. In my opinion, it cannot be reached sooner or later. At most, insects can reach sexual maturity earlier.

L315: “pH” – normal font, not italic.

L326: Whether it has been proven that these bacteria are symbionts of Dysdercus? If not - correct.

L426: “unexpected”? Is this effect really unexpected? Considering the mechanism of action of Bacillus thuringiensis toxins? Correct.

L432: as above (correct “unexpected”).

Author Response

Response to Reviewer 2 Comments

We are grateful for the comments and suggestions of reviewer 2, which undoubtedly contributed to enrich our manuscript (#2104795). The comments and the authors' response are listed below.

Point 1: L22: “evolutionary history” - this is an overreaching statement.

Response 1: We decided to rewrite line L22 to reduce its scope and make its meaning true to the message we want to convey in our manuscript.

Point 2: L92: insert a space before GM.

Response 2: Ok, we added a space, but we reworded the paragraph to respond to reviewer 1.

Point 3: L113: replace “sterile” to “sterilized”.

Response 3: Ok, we replace the word

Point 4: L144: replace “statistical” to “statistically”.

Response 4: Ok, we replace the word

Point 5: Figure 1: If possible - change the type of the chart so that it is readable as a black and white printout. Enlarge the fonts.

Response 5: We decided to keep the color format of Figure 1 and increased the font size to improve legibility.

Point 6: Table 2: change “Df” to “df”.

Response 6: Ok, we replace the word.

Point 7: Table 3: genus and species names of bacteria should be in italics.

Response 7: Ok, we highlight the bacterial genera and species mentioned throughout the manuscript, including Tabla 3.

Point 8: Table 4: insert commas between genus names of bacteria.

Response 8: Ok, we insert commas between the bacterial genus names

Point 9: L288: remove “exciting, don't judge your own research results.

Response 9: Ok, we replace the word.

Point 10: L288: change “mutualistic” to “symbionts”.

Response 10: Ok, we replace the word.

Point 11: L311: “reproductive age earlier” - age is measurable. In my opinion, it cannot be reached sooner or later. At most, insects can reach sexual maturity earlier.

Response 11:We agree with the reviewer's observation, we decided to modify the sentence by "reach sexual maturity earlier".

Point 12: L315: “pH” – normal font, not italic.

Response 12: We substitute the word in italics.

Point 13: L326: Whether it has been proven that these bacteria are symbionts of Dysdercus? If not - correct.

Yes, they have been reported before. We decided to rewrite line L326 for better understanding and clarity.

In the study by Sudakaran et al., 2015, they investigated whether the acquisition of vitamin-providing symbionts allowed their hosts to diversify into nutritionally unbalanced and chemically well-defended malvaceous plant seeds as a food source. Their results indicate that vitamin-providing Actinobacteria (Coriobacterium and Gordonibacter), as well as vitamin-providing Firmicutes (Clostridium) and Proteobacteria (Klebsiella), are widespread in the Pyrrhocoridae, but absent in the sister family Largidae and other non-autochthonous taxa.

Sudakaran, S.; Retz, F.; Kikuchi, Y.; Kost, C.; Kaltenpoth, M. Evolutionary Transition in Symbiotic Syndromes Enabled Diversification of Phytophagous Insects on an Imbalanced Diet. ISME J. 2015, 9, 2587–2604, doi:10.1038/ismej.2015.75.

Point 14: L426: “unexpected”? Is this effect really unexpected? Considering the mechanism of action of Bacillus thuringiensis toxins? Correct.

Response 14: Ok, we decided to remove the word "unexpected", to keep only with the term: "residual effect" since the main focus of the manuscript is to gather information on residual effects of the presence of transgenes in natural ecosystems. We change this terminology in all manuscript.

Point 15: L432: as above (correct “unexpected”)..

Response 15: We decided to remove the word "unexpected", to keep only with the term: "residual effect" since the main focus of the manuscript is to gather information on residual effects of the presence of transgenes in natural ecosystems. We change this terminology in all manuscript.